# Exploring Consumer Behavior and Preferences in Welfare-Friendly Pork Breeding: A Multivariate Analysis

**DOI:** 10.3390/foods12163014

**Published:** 2023-08-10

**Authors:** Michela Pugliese, Annalisa Previti, Angelina De Pascale, Angela Alibrandi, Agata Zirilli, Vito Biondi, Annamaria Passantino, Salvatore Monti, Carlo Giannetto, Maurizio Lanfranchi

**Affiliations:** 1Department of Veterinary Sciences, University of Messina, Via Umberto Palatucci, 98168 Messina, Italy; michela.pugliese@unime.it (M.P.); annalisa.previti@yahoo.it (A.P.); vito.biondi@unime.it (V.B.); annamaria.passantino@unime.it (A.P.); smonti@unime.it (S.M.); 2Department of Economics, University of Messina, Via dei Verdi 75, 98122 Messina, Italy; adepascale@unime.it (A.D.P.); angela.alibrandi@unime.it (A.A.); agata.zirilli@unime.it (A.Z.); mlanfranchi@unime.it (M.L.)

**Keywords:** pork consumers, purchasing habits, choices, welfare-friendly livestock, animal welfare

## Abstract

This study investigates consumer behavior and interest in “welfare-friendly” forms of pork production, considering the growing presence of animal welfare-focused breeding practices. The aim is to outline the typical profile of pork consumers and identify the key attributes influencing their purchasing decisions. A survey was conducted on a sample of 286 individuals after excluding those who only consumed beef and/or poultry or identified as vegetarians/vegans. Regression coefficients (b), 95% Confidence Intervals (95% C.I.), and *p*-values were reported for univariate and multivariate models. Statistical significance was determined at *p* < 0.05 (indicated in bold). The findings indicate that younger participants show greater sensitivity towards consuming meat raised using welfare-friendly methods, raising considerations about the age composition of the sample. The research’s originality lies in evaluating consumer interest in pork raised with animal welfare-respecting techniques. The use of appropriate statistical tools, such as multivariate and multilayer models, allows effective solutions for multidimensional hypothesis testing problems in non-parametric permutation inference.

## 1. Introduction

Pork is one of the most used meats in the world; in fact, in April 2022, about 778.64 million pigs were registered worldwide [1]. In particular, China is the leading pork producer worldwide, while the EU is the top exporter country, with about 3.75 million metric tons of pork shipments in the most recent year [1]. Intensification and specialization of techniques used in the current intensive rearing systems lead to higher economic efficiency of production, but they cause serious ecological problems as well as problems related to animal welfare, herd health and food safety [2,3]. On the other hand, consumers have become more scrupulous in understanding how their food is produced, so the controls have increased regarding intensified food animal production methods in order to obtain an optimal food safety level [2]. Among the primary concerns associated with intensive pig production systems, there is a demonstrated correlation between various sources of stress in farms and the resulting welfare problems. These stressors can arise from factors such as inadequate housing conditions, suboptimal stockmanship practices and/or unfavorable environmental factors. Consequently, pigs may experience chronic hunger, painful mutilations, early weaning, high stocking density, successive social regrouping and increased susceptibility to pathogens [4]. For this reason, these kinds of farms rely on the use of antibiotics for the treatment of infection, contributing to the global issue of antimicrobial resistance; this requires global efforts on the prudent use of antibiotics in response to the emergence of antimicrobial resistance (AMR) in farming [5]. The global consumption of veterinary antimicrobials, in fact, is projected to increase 11.5% by 2030, despite several countries having banned the use of antibiotics such as growth promoters and/or for preventive uses and having limited some active principles for human use only [3]. On the other hand, a high level of farm animal welfare is a guarantee of good health, both of the animal and, consequently, of the consumer, for example, through the elimination of antibiotics or other used drugs [6].

This issue has necessitated greater regulation of antimicrobial use in the veterinary sector, as specified by Regulation (EU) 2019/6. Despite the legislation restricting AMU in livestock production, most notably in the EU, the use of veterinary antibiotics in the world, in fact, is still irregular. For example, it is reported that in Germany, pigs with an expectation of 200 days of life received antimicrobials for 48.5 days, and in Brazil, pigs received an average of seven different antibiotic active principles during 73.7% of their life [3].

In order to control the spread of antibiotic resistance and to protect public health, it is necessary to reduce antimicrobic use through their prudent use and to foster good husbandry practices, including improving animal welfare [7,8,9]. Detailed rules applying to prudent use can also be found in Commission Notice 2015/C 299/04, regarding “Guidelines for the prudent use of antimicrobials in veterinary medicine”.

The ‘One Health’ concept affirms that human health is wholly linked with animal health [10]. This leads to another one emerging, the ‘One Welfare’ paradigm, which shows human welfare and animal welfare as interdependent and bound to the state of the ecosystems in which they exist [11]. Hence, a “One Welfare” issue affects human, animal and environmental welfare and highlights the fragility of intensive high-throughput livestock production systems [11]. The COVID-19 pandemic has also contributed to making consumers more aware of this link between health, ecosystems, supply chains, consumption patterns and the environment, and it has led to increased awareness amongst consumers towards One Health concepts and animal welfare information on food labels [12,13,14].

Despite the increasing attention to animal welfare and the sensitivity of European consumers towards this issue, in many countries, farmed pigs routinely undergo a set of invasive procedures during their lives, such as surgical castration, tail docking and teeth clipping without appropriate pain management [15]. For example, relating to surgical castration, it is performed in order to reduce aggressiveness and eliminate “boar taint”, an unpleasant odor of the meat that results from the accumulation of skatole and androsterone after puberty. One of the alternatives is immunocastration by an anti-GnRH vaccine, which has been shown to prevent boar taint [15].

Public opinion pressure is increasing for food-producing animal origin to be more consistent with sustainability and animal welfare goals; in fact, farm animal welfare (FAW) is the main demand from society and, consequently, politics [16]. Therefore, farm animal husbandry systems must be adapted to animal welfare in line with social demands and economic efficiency [17]. However, the actions aimed at improving animal welfare not only have a positive impact by enhancing pig production but also come with certain costs and uncertainties. These costs include learning and opportunity costs, as pig farmers may need to make changes to their housing and management system, potentially reducing their monetary benefits and their willingness to adopt a new system. Additionally, uncertainties such as financing, a lack of political decisions and environmental protection issues related to building permits further unsettle pig farmers [17]. A better understanding of the factors that condition the consumers’ perceptions of the FAW is an essential step toward enhanced sustainability and social responsibility in modern food production systems because these factors can influence the behavior at the time of food purchase [2].

European consumers pay more attention to FAW standards, and considerations about this matter have begun to feature strongly in their decision making [18]; nevertheless, they face a lot of challenges when purchasing animal-friendly products, which may lead to attitude–behavior discrepancies (ABDs) [18]. This gap shows a possible mismatch between a person’s role as a citizen or as a consumer because their perceptions and priorities may diverge when considering societal issues versus personal consumption choices, and it may be explained with a multitude of reasons such as difficulty processing information, abrogation of responsibility to others, affordability, availability, etc. Understanding and addressing these discrepancies is crucial for effectively promoting sustainable behaviors and aligning individuals’ actions with their stated attitudes and values [19,20].

In many countries, consumers are willing to accept that it is ethically acceptable to eat meat provided the animals enjoy a good level of welfare, and some studies reveal that consumer willingness to pay (WTP) was higher for products obtained using animal-friendly farming techniques [21,22,23]. In particular, the citizens’ WTP is an average 5% extra for pork from outdoor-raised pigs, and one-fifth of consumers claim to be willing to pay 20% extra [24,25].

This study aims to explore the knowledge of consumers toward farm pig welfare and management in order to define their standard profile, studying the relationship between demographic variables and animal welfare concerns. These attitudes can condition the choice at the time of purchase and impact farmer management strategies.

## 2. Materials and Methods

### 2.1. Ethical Standards

First, human participants read some information about the study and gave their consent to take part by signing a consent before data collection, based on the Regulation (EU) 2016/679 regarding the processing of personal data and Italian legislation (Legislative Decree no. 165 of 30 March 2001 and the Legislative Decree. no. 33 of 14 March 2013).

### 2.2. Consumer Survey Design

The survey was conducted within several municipalities in the Messina province (town in the region of Sicily, southern Italy), near very busy places (supermarkets, main squares, schools, etc.) in a random way in order to ensure the presence of different types of individuals. The data were collected through an ad hoc questionnaire between September and December 2022.

A total of 450 individuals purchasing and consuming pork meat (and derivatives) compiled the questionnaire in paper format.

The questionnaire (Appendix A), which was anonymous, was directly distributed with the face-to-face method. It was divided into three different sections. The first section included information related to socio-demographical characteristics such as gender, age, educational qualification, income range, number of family members and religious beliefs (yes/no).

In the second part, the consumers were asked to indicate, using a scale of 0 (not important to 10 (very important), how important the following factors were in determining why they would purchase or eat pork: product price, company brand, organoleptic characteristics, breeding techniques, Italian origin of the pigs, location of the farm, DOC/IGP brand, organic farming methods.

The third section contained questions relating to the consumer’s sensitivity towards animal welfare (yes/no), to knowledge of mutilating practices permitted by legislation (yes/no), to the belief that the application of mutilating practices is right (yes/no), favorable to castration (yes with traditional surgery, yes with anesthesia or no with preference to alternative methods such as immunocastration). The last questions referred to the consumer’s preference for sustainable farms (where sustainability was defined by Ruckli et al. [26] as “the ability to make development sustainable to ensure that it meets the needs of the present without compromising the ability of future generations to meet their own needs” and a holistic approach that seeks to create a harmonious balance between environmental protection, economic prosperity, social well-being and animal welfare ensuring a better future for both current and future generations) compared to non-sustainable farms (yes/no) and to the perception of adequate publicity of sustainable farms (yes/no).

After eliminating respondents that did not meet the requirements, such as those who ate only beef and/or poultry and/or were vegetarians/vegans, the final sample included 286 individuals.

### 2.3. Statistical Analysis

First, the dataset was cleaned to verify that participants met the inclusion criteria of being a pork-eater.

In order to summarize the single factors of choice and consumption of pork meat, we realized a normalized synthesis index (indicated with Is), which varied from 0 to 1. It is given by the sum of the scores of the single eight factors divided by the maximum theoretical value, i.e., 80, which corresponds to the score we would obtain if all the answers were associated with the maximum score of 10.

The numerical variables (age and Is) were expressed as mean, standard deviation (S.D.) and interquartile range (Q1–Q3), categorical variables (sex, educational qualification, family components, income range, religious believers, sensitivity to animal welfare, knowledge of the mutilations, belief that the application of mutilating practice is right, favorable to castration, preference for sustainable farms, adequate publicity of sustainable) such as absolute frequencies and percentages.

A radar chart was generated in order to display, for each indicator of importance (ranging between 0 (not important) and 10 (very important)), the means of scores expressed by pork consumers.

In order to identify the factors that significantly influence this synthesis index Is (outcome of interest), univariate and multivariate linear regression models were estimated [27]. The tested covariates were the following: sex, age, educational qualification, income range, number of family components, religious believers, sensitivity to animal welfare, knowledge of mutilations permitted by law, justifiability of mutilating practices, favorable to castration, preference for sustainable farms, adequate publicity of sustainable farms. The results of univariate and multivariate models were reported as regression coefficient (b), 95% Confidence Interval (95% C.I.) and *p*-value.

We considered statistically significant all *p*-values lower than 0.05 (reported in bold).

Statistical analyses were performed using SPSS for Windows Package, version 22.0.

## 3. Results

Table 1 shows absolute frequencies and percentages for categorical variables. Examining the descriptive statistics, the gender of the respondents was equally represented. The sample was mainly composed of individuals with a high level of education (61.9% high school and 33% degree), with a number of family members equal to 4, an income between EUR 10,000 and 49,999, mostly religious, sensitive to animal welfare, mostly in favor of castration as long as it is carried out with anesthesia and with a preference for sustainable farms, for which, however, they do not find adequate publicity.

Descriptive statistics for numerical variables were reported in Table 2.

Examining the distribution of the sample according to gender, there was an equal distribution (48.6% male and 51.4% female). Regarding educational qualification, as many as 91.9% of respondents had high school, while 33.9% had a degree. Mostly the number of household members of respondents was four (38.5%); about the income, a fairly high percentage (47.6%) belonged to an income range between EUR 10,000 and 29,999. Predominantly, the sample consisted of individuals with religious beliefs (72.4%), and they declared themselves to be sensitive to animal welfare at a percentage of 89.2%. The majority of the samples stated that they did not know the mutilating practices allowed by law (60.1%); 71.3% of the respondents believed that the mutilating practices are not right, and 65.8% were favorable to castration with anesthesia. Finally, 84.3% expressed a preference for sustainable farms, and 91.6% stated that the publicity for sustainable farms is not adequate.

Figure 1 is a radar chart formed by eight axes, representing the eight indicators of importance: product price, corporate branding, organoleptic characteristics, breeding techniques, Italian origin of pigs, farm location, DOC/IPG brand and organic farming method) (ranged between 0 and 10). It shows the mean of each score deriving from the judgments of importance expressed by consumers of pork meat. In this radar chart, higher average scores (expressing greater importance) are identified by points further out, minor scores (minor importance) are identified by dots placed closer to the center; so, we can note that consumers attributed greater importance to items such as “Italian origin of pigs” (mean 8.0 ± 2.4) and “organic farming methods” (mean 7.5 ± 2.5), while minor importance was attributed to “corporate branding” (mean 5.2 ± 2.9) and “product price” (mean 6.0 ± 2.5).

The results of univariate and multivariate models are reported in Table 3.

The estimation of the regression models allowed us to identify the factors that significantly influence the summary index of importance (Is); in particular, in both the univariate and multivariate approaches, four significant covariates were identified: sex (with males expressing higher levels of importance), religious belief, sensitivity towards animal welfare and the preference for sustainable farms, which promote animal welfare.

## 4. Discussion

This study provides information on the standard profile of the pork consumer, shedding light on their characteristics, preferences and behaviors. Furthermore, by highlighting the characteristics that are pre-eminent for consumers at the point of purchase, the study aims to provide valuable insights and guidance for pork producers, marketers and retailers in developing effective strategies to meet consumer demands and enhance their overall satisfaction [28,29].

These results suggest a reflection concerning the age of the respondents’ sample, meaning that the youngest participants appear more sensible to the consumption of meat raised with animal-welfare-friendly methods. Regarding this age factor, several studies reported conflicting opinions. According to Vargas-Bello-Pérez et al. (2017), age does not affect the consumers’ attitude toward animal welfare, although, in general, the people’s sensitivity depends on what they think in their different roles as citizens and as consumers due to ABDs [29,30,31,32]. On the other hand, several researchers have highlighted that the youngest people have a more positive attitude toward animals, showing an increased interest in their care and well-being, due probably to an increased opportunity to hear about “animal welfare” [33,34,35]. More empathetic responses were generally found in younger participants, although this did not always translate into WTP more for higher-welfare animal products. Younger consumers are more likely to support ethical considerations in their purchasing decisions, including those that prioritize animal welfare. They view animals as sentient beings deserving of compassion and respect, and they want to ensure that their purchases do not contribute to animal suffering [33]. According to our results, the youngest people often believe that all of society is responsible for animal care in order to improve food safety, human health and the quality of animal products [36,37,38]. Cornish et al. 2020 [39]—using an adapted version of the Animal Attitude Scale which measured the consumers’ pro-welfare attitudes—showed that the empathy measures (Animal Empathy Score) are highest in younger participants than in older. This could be linked to a better consumer understanding of animal welfare due to the development of current legislation on the issue and the concept of animal welfare science. In fact, the matter has developed prevalently over the last 30 years, but it has become a fundamental scientific basis on which important political decisions are made [40]. It is reasonable that younger people, experiencing first-hand the evolution of concern for animal welfare, are more susceptible than the elderly, who have more rooted ideologies [40,41].

To understand the main sources of information used by the general public and their prevalence relating to demographic variables like age, it is fundamental to individuate whom consumers look to for guidance on animal welfare topics. Based on ISTAT [42] and Eurostat [43], young people are the most common users of the internet as an information source, and the youngest use PCs and smartphones in combination or exclusively via smartphones, while people aged 65 and over represent the highest share of those who access it exclusively through the PC. In the Authors’ opinion, younger generations have probably grown up in an era of heightened awareness because they are more exposed to information through social media, documentaries and educational campaigns, which has led to a greater understanding of the impact of their purchasing choices on animals and the planet. However, the COVID-19 pandemic has contributed to changing consumers’ daily routines; being confined to their homes, many people increased their overall internet use, having more time available [44,45,46].

Furthermore, our data shows that consumers attribute greater importance to items such as “Italian-origin pigs”. This definition includes Protected Designations of Origin (PDO) Italian pork products, among which are well-known cured hams such as Parma ham and San Daniele ham, appreciated all over the world, that constitute the majority of Italian pork manufacturing [47,48]. Based on data collected by the consortium in 2020 [49], its production has reached a revenue from consumer sales of EUR 1.500 million and exports amounting to 29% of the 8.7 million hams produced. Moreover, PDO products are considered by consumers to be more responsive to animal welfare concerns, so consumers expect special quality characteristics from PDO products, including ethical propriety [50,51,52]. Nevertheless, to obtain the organoleptic qualities of these typical Italian products, it is necessary that the pigs are castrated to prevent “boar taint” in meat. According to Directive 120/2008/EC, surgical castration is allowed, such as tail docking, and it generally effectuates within the first week of life, without anesthesia and/or analgesia [53,54], causing chronic pain and distress in piglets (EFSA) [55]. For this reason, many other castration techniques have been recommended since 2010 by the European Declaration. Castration is still an unavoidable mutilating practice for typical Italian products such as hams. Consequently, immunocastration is a viable alternative, but due to their longer lifespan and high weight to be reached before slaughter, Italian heavy pigs need an additional dose of vaccine, which results in higher costs and greater procedures in animals that are difficult to handle [56,57]. However, the consumers’ WTP—a premium price to buy meat from immunocastrated pigs—shows an optimal acceptance of this kind of technique, which is perceived positively with a relatively low level of risk perception [58,59].

This study also revealed the prevalence of consumer attributes of greater importance in organic farming methods. In fact, the FAW concept is considered an integrating part of the internationally organic principles on health, ecology, fairness and care, reflecting the expectation of consumers and farmers. As reported in several studies, consumers assume that organic livestock enjoys high standards of FAW [60,61,62,63,64]. This assumption is based on the belief that organic farms prioritize the well-being of their livestock with specific requirements for animal welfare, such as access to outdoor areas and space to move and exhibit natural behaviors and restrictions on the use of antibiotics and hormones [60,61,62,63,64].

The COVID-19 pandemic has highlighted, in general, worldwide, and especially in Italy, a deep problem in the pre-existing conditions within the food production system, which is linked to a decrease in public health and greater environmental degradation related to increased carbon emissions [65,66]. This pandemic has led to rapid changes in daily life, including a different approach to consumers’ eating habits, also influenced by mass media and individual social or emotional states. This new viewpoint had many effects on the food purchased and has brought better attention to health and the environment, as well as production methods, ethical and social responsibility, in order to improve the required ecological transition [67,68]. According to Russo et al. (2021), in the design of marketing strategies in response to changes in demand following the COVID-19 emergency, it is crucial to consider both psychological pressure and consumer characteristics. In particular, the firms targeting young consumers should anticipate short-lived behavioral changes, as this demographic may be more prone to quickly revert back to their pre-pandemic consumption patterns, while older consumers are more likely to adopt and sustain changes in the long run, making them a potentially more receptive target market for long-term marketing strategies [69].

The current study has limitations due to a recruitment bias that could occur in that participants who have had an existing interest in the topic may be more predisposed to take part in the studies. Nevertheless, the survey reflected the views of a large sample representative of data demographics, thus providing rigor to the findings.

## 5. Conclusions

The results of our survey can be traced to Southern Italy and, particularly, to the city of Messina (Sicily).

Consumers in Western countries have acquired an increased interest in FAW, and, therefore, for them, the price is not the only determining factor in purchases of animal-based food. In fact, Italian consumers give greater importance to products obtained with sustainable breeding techniques and by national farms, perceiving greater attention on FAW in organic farms.

Certainly, the COVID-19 pandemic has sensitized consumers’ opinions on issues relating to animal, human and environmental health, directing them to a “One-Health” perspective even more.

It would be interesting to apply a certification system based on the principles of “One Health”, similar to what the US already does for other meats, in order to guarantee that pork is produced under a transparent program of best responsible care practices such as the modern consumer would like [39,68].

## Figures and Tables

**Figure 1 foods-12-03014-f001:**
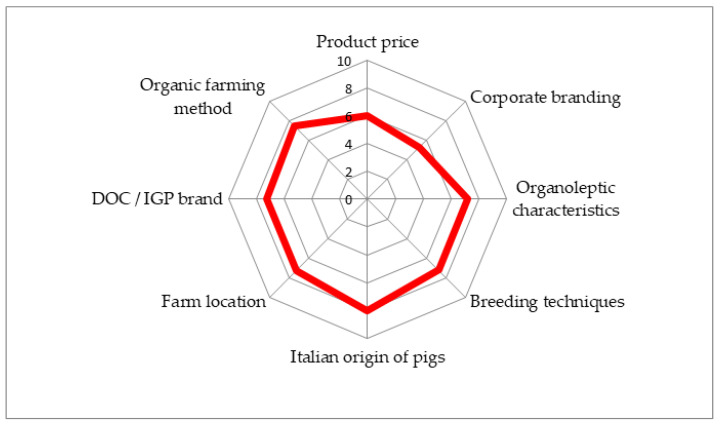
Radar chart on importance attributed to different factors of choice and consumption.

**Table 1 foods-12-03014-t001:** Absolute frequencies and percentages for categorical variables.

SEX	
M	139 (48.6%)
F	147 (51.4%)
EDUCATIONAL QUALIFICATION	
Primary school	3 (1%)
Middle school	9 (3.1%)
High school	177 (61.9%)
Degree	97 (33.9%)
FAMILY COMPONENTS	
1	20 (7.0%)
2	31 (10.8%)
3	73 (25.5%)
4	110 (38.5%)
>4	52 (18.2%)
INCOME RANGE	
<EUR 9.999	49 (17.1%)
EUR 10.000–19.999	72 (25.2%)
EUR 20.000–29.999	64 (22.4%)
EUR 30.000–49.999	58 (20.3%)
EUR 50.000–69.999	19 (6.6%)
EUR 70.000–99.999	13 (4.5%)
>EUR 100.000	11 (3.8%)
RELIGIOUS BELIEVERS	
Yes	207 (72.4%)
No	79 (27.6%)
SENSITIVITY TO ANIMAL WELFARE	
Yes	255 (89.2%)
No	31 (10.8%)
KNOWLEDGE OF THE MUTILATIONS PERMITTED BY LAW	
Yes	114 (39.9%)
No	172 (60.1%)
BELIEF THAT THE APPLICATION OF MUTILATING PRACTICES IS RIGHT	
Yes	82 (28.7%)
No	204 (71.3%)
FAVORABLE TO CASTRATION	
Yes (with traditional surgery)	52 (53.1%)
Yes (with anesthesia)	100 (65.8%)
No (preference for alternative methods such as immunocastration)	134 (46.9%)
PREFERENCE FOR SUSTAINABLE FARMS	
Yes	241 (84.3%)
No	45 (15.7%)
ADEQUATE PUBLICITY OF SUSTAINABLE FARMS	
Yes	24 (8.4%)
No	262 (91.6%)

**Table 2 foods-12-03014-t002:** Descriptive statistics for numerical variables.

Descriptive Statistics	Mean	S.D.	Q1–Q3
Age	29.56	11.95	20.75–35.00
Is	0.70	0.17	0.60–0.83

**Table 3 foods-12-03014-t003:** Univariate and multivariate linear regression model for Is.

	Univariate			Multivariate		
Covariates	b	95% C.I.	*p*-Value	b	95% C.I.	*p*-Value
Sex (M vs. F)	0.043	0.005; 0.082	0.027	0.045	0.007; 0.084	0.022
Age	0.001	−0.001; 0.003	0.224	0.001	−0.001; 0.002	0.425
Educational qualification	0.030	0.009; 0.069	0.128	−0.010	−0.045; 0.025	0.584
Income range	−0.002	−0.015; 0.010	0.724	−0.003	−0.015; 0.010	0.689
Family components	−0.013	−0.029; 0.003	0.114	−0.009	−0.026; 0.008	0.293
Religious believers (yes/no)	0.040	0.001; 0.080	0.035	0.039	0.004; 0.083	0.045
Sensitivity to animal welfare (yes/no)	0.097	0.035; 0.159	0.002	0.102	0.038; 0.165	0.002
Knowledge of mutilations permitted by law (yes/no)	0.010	−0.030; 0.050	0.608	0.008	−0.034; 0.050	0.717
Justifiability of mutilating practices (yes/no)	0.036	−0.007; 0.079	0.099	0.048	−0.002; 0.097	0.078
Favorable to castration (yes/no)	−0.003	−0.043; 0.036	0.870	−0.026	−0.071; 0.020	0.263
Preference for sustainable farms	0.048	0.002; 0.094	0.041	0.048	0.002; 0.094	0.039
Publicity of sustainable farms	0.001	−0.069; 0.072	0.970	−0.003	−0.074; 0.068	0.938

## Data Availability

Data are contained within the article.

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
