# Peer review of "Exploring Consumer Behavior and Preferences in Welfare-Friendly Pork Breeding: A Multivariate Analysis"

_foods, 2023, doi:10.3390/foods12163014_

Round 1

Reviewer 1 Report

The aim of this study is relevant to the results and conclusions obtained. To define the profile of pork consumers and to identify the attributes that influence the consumption and purchase decision. 

The study is important because it presents information that can apparently be perceived from the population, although what is relevant in this study is that it uses appropriate statistical tools to analyze the information from non-parametric data. This makes it possible to observe and point out consumer profile and attributes in terms of considering breeding in compliance with animal welfare guidelines. And compare the trends that have been observed in other countries.

Line 184: review writing 

Author Response

The aim of this study is relevant to the results and conclusions obtained. To define the profile of pork consumers and to identify the attributes that influence the consumption and purchase decision.

Authors’ response: Thank you. We sincerely appreciate the reviewer's consideration of our work.

The study is important because it presents information that can apparently be perceived from the population, although what is relevant in this study is that it uses appropriate statistical tools to analyze the information from non-parametric data. This makes it possible to observe and point out consumer profile and attributes in terms of considering breeding in compliance with animal welfare guidelines. And compare the trends that have been observed in other countries.

Authors’ response: Thank you. We sincerely appreciate the reviewer's consideration of our work.

Reviewer 2 Report

Line 29: newer statistical data could be presented

Line 32: can you provide more proof for statement: “they cause serious ecological problems“, because papers that are cited do not provide such information.

Line 33-37: Scientific evidence is needed to support such statement because paper that is cited deals with citizens’ perception of modern pig production in Germany not with intensive pig production issues.

Line 37-38: Statement derived from article 5 (Willingness to pay and moral stance: The case of farm animal welfare in Germany) is considered as not justified because there is no evidence presented nor the aim of that article is to deal with „farm rely on the use of antibiotics for preventive purposes contributing to the global issue of antimicrobial resistance“. Even more, preventive use of antibiotics is prohibited in EU.

Line 38-41: Statement derived from article 6 (Consumers’ Preference for the Consumption of the Fresh Black Slavonian Pig’s Meat) is considered as not justified because there is no evidence presented nor the aim of that article is in line with the statement.

Line 42-45: Assumptive statement: there were no scientific and experimental evidence presented in Introduction that can be derived to those statements.

Line 46-51: One Health and One Welfare concepts should be better explained, because supportive articles (7 and 8) do not provide such an information. Further more, statement derived from article 8 does not corresponds with article’s results nor aim (“the aim of the present study to find different consumer groups that differ in their attitudes and their shopping behaviour and to derive target groups for AWPs”).

Line 51-54: Statement given in manuscript does not corresponds with articles’ aims, results nor conclusions (article 9, 10, and 11)

Line 55-61: unclear how this section is connected to the title and the aim.

Line 69-70: do you have an evidence for the statement “Despite the European consumers pay more attention to FAW standards and considerations about this matter have begun to feature strongly in their decision-making“, because article that is refeered to is not of that matter (18. Consumers’ Perception and Preference for the 317 Consumption of Wild Game Meat among Adults in Poland).

Line 75: Cannot find a reason to include article 20 (Foods for plant-based diets: Challenges and innovations) in that section; need to revise.

Line 82-83: aim of the study and the title are not related; there is no sustainable pork breeding in aim, but pig welfare and management; title should be changed

An example of the questionnaire should be given as figure.

Line 110-116: what was the reason to include such questions as yes/no question? Seems that it is harder to process binary data with 0 and 1 value in proper way.

Line 114: description of sustainable farm as one that is oriented towards animal welfare is ambitious and incorrect; please be advised to see the special issue: Sustainability | Special Issue : Sustainable Pig Production (mdpi.com)

Line 158: explanation and importance of the results in Table 2 is missing.

Line 170: company brand and Corporate branding in figure 1 are different labels.

Line 208-217: elaborated section was not a part of research, and also why was made a connection with COVID?

Line 231-239: immunocastration was not elaborated in questionnaire-why is it elaborated here? You have not elaborated data where more than half of respondents are for surgical castration.

Line 240-244: how was the relationship between organic farming and animal welfare tested? Consumers that you referring to are not from your research.

Line 245-252: COVID issues were not presented to respondents – why elaborate on that?

Line 255: having 286 respondents over a population in Italy of about 60.000.000 can not be considered as a large sample for representation.

Language writing in scientific appropriate manner is needed, especially Materials and methods section (avoid writing such as we realized…)

Author Response

Line 29: newer statistical data could be presented  

Authors’ response: Thank you for your suggestion. We have provide to report newer statistical data

Line 32: can you provide more proof for statement: “they cause serious ecological problems“, because papers that are cited do not provide such information.

Authors’ response: Thank you for your suggestion. We have provide to change the references. There was a mistake with the references.

Line 33-37: Scientific evidence is needed to support such statement because paper that is cited deals with citizens’ perception of modern pig production in Germany not with intensive pig production issues.

Authors’ response: Thank you for your suggestion. We have provide to change the references. There was a mistake with the references.

Line 37-38: Statement derived from article 5 (Willingness to pay and moral stance: The case of farm animal welfare in Germany) is considered as not justified because there is no evidence presented nor the aim of that article is to deal with „farm rely on the use of antibiotics for preventive purposes contributing to the global issue of antimicrobial resistance“. Even more, preventive use of antibiotics is prohibited in EU.

Authors’ response: Thank you for your suggestion. We have provide to change the references. There was a mistake with the references.

Line 38-41: Statement derived from article 6 (Consumers’ Preference for the Consumption of the Fresh Black Slavonian Pig’s Meat) is considered as not justified because there is no evidence presented nor the aim of that article is in line with the statement.  

Authors’ response: Thank you for your suggestion. We have provide to change the reference. There was a mistake with the references.

Line 42-45: Assumptive statement: there were no scientific and experimental evidence presented in Introduction that can be derived to those statements.

Authors’ response: Thank you for your suggestion. We have provide to change the references. There was a mistake with the references.

Line 46-51: One Health and One Welfare concepts should be better explained, because supportive articles (7 and 8) do not provide such an information. Further more, statement derived from article 8 does not corresponds with article’s results nor aim (“the aim of the present study to find different consumer groups that differ in their attitudes and their shopping behaviour and to derive target groups for AWPs”).

Authors’ response Thank you for your suggestion. We have provide to change the references. There was a mistake with the references.

Line 51-54: Statement given in manuscript does not corresponds with articles’ aims, results nor conclusions (article 9, 10, and 11)

Authors’ response: Thank you for your suggestion. We have provide to change the references. There was a mistake with the references.

Line 55-61: unclear how this section is connected to the title and the aim.

 Authors’ response: We have modified this section.

Line 69-70: do you have an evidence for the statement “Despite the European consumers pay more attention to FAW standards and considerations about this matter have begun to feature strongly in their decision-making“, because article that is refeered to is not of that matter (18. Consumers’ Perception and Preference for the 317 Consumption of Wild Game Meat among Adults in Poland).

Authors’ response: Thank you for your suggestion. We have provide to change the references.

Line 75: Cannot find a reason to include article 20 (Foods for plant-based diets: Challenges and innovations) in that section; need to revise.

Authors’ response: Thank you for your suggestion. We have provide to change the references

Line 82-83: aim of the study and the title are not related; there is no sustainable pork breeding in aim, but pig welfare and management; title should be changed

Authors’ response: We have modified the title

An example of the questionnaire should be given as figure.

Authors’ response: our questionnaire was attached as “supplementary matherial.”

Line 110-116: what was the reason to include such questions as yes/no question? Seems that it is harder to process binary data with 0 and 1 value in proper way.

Authors’ response: In third section of the questionnaire, we actually wanted to survey data about consumer sensitivity to animal welfare. We avoided detecting qualitative variables, for which it is only possible to statistically investigate associations. We preferred to detect single dichotomous variables (yes/no) because these are amenable to being tested in a regression model and thus it is possible to assess their effect on the response variable (outcome of interest).

Line 114: description of sustainable farm as one that is oriented towards animal welfare is ambitious and incorrect; please be advised to see the special issue: Sustainability | Special Issue : Sustainable Pig Production (mdpi.com)

Authors’ response: Thank you for your suggestion. We have corrected the definition on the basis of the suggested special issue.

Line 158: explanation and importance of the results in Table 2 is missing.

Authors’ response:  We thank the referee for the valuable suggestion to improve the overall quality of our manuscript. In the revised version of our article, we explained the descriptive statistics shown in Table 2. In particular we added the following sentence “ Examining the distribution of the sample according to gender, there is an equal distribution (48.6% male and 51.4%female). Regarding educational qualification, as many as 91.9% of respondents have high school, while 33.9% are degree. Mostly the number of household members of respondents is 4 (38.5%); about the income, a fairly high percentual (47.6%) belongs to an income range between 10,000€ and 29,999€. Predominantly, the sample consists of individuals who have a religious belief (72.4%) and declare themselves to be sensitive to animal welfare at a percentage of 89.2%. The majority of the sample states that they do not know the mutilating practices allowed by law (60.1%), the 71.3% of the respondents believe that the applications of mutilating practices in not right and the 65.8% is favorable to castration with anesthesia. Finally, the 84.3% express preference for sustainable farms and the 91.6% states that the publicity for sustainable farms is not adequate.”

Line 170: company brand and Corporate branding in figure 1 are different labels.

Authors’ response: We have reported the sentence “corporate branding” as in figure 1.

Line 208-217: elaborated section was not a part of research, and also why was made a connection with COVID?

Authors’ response: Covid has profoundly changed consumer eating habits, so we have emphasized this important aspect.

Line 231-239: immunocastration was not elaborated in questionnaire-why is it elaborated here? You have not elaborated data where more than half of respondents are for surgical castration.

Authors’ response: Immunocastration was considered. In fact, in materials and methods is explained that the third section contained questions also relating the castration [the favorable to castration (yes with traditional surgery, yes with anesthesia or no)]. Relating to no, we have better specified addeding “with preference to alternative methods such as immunocastration” (See line 116-117). The same in table 1.

Line 240-244: how was the relationship between organic farming and animal welfare tested? Consumers that you referring to are not from your research.

Authors’ response: The relationship between organic farming and animal welfare is explained in lines 262-263. In organic farming methods the concept of FAW (that we treat) is considered an integrating part in line with internationally agreed biological principles of health and ecology. We have modified the sentence, because the consumers that we referring are not from our survey.

Line 245-252: COVID issues were not presented to respondents – why elaborate on that?

Authors’ response: We are aware that COVID issues were not presented to respondents; however we cannot fail to highlight the role of Covid, which has certainly changed the consumer's eating habits.

Line 255: having 286 respondents over a population in Italy of about 60.000.000 can not be considered as a large sample for representation.

Authors’ response: In our manuscript, section 2, subsection 2.2 we specified that “The survey was conducted within several municipalities in the province of Messina (cities in the region of Sicily, southern Italy).” The reference population is 218,187 subjects. For this reason our sample can be considered as a large sample for representation and it is suitable to ensure the extensibility of the results.

Reviewer 3 Report

-As pointed out by the authors, there is a high likelihood of sampling bias based on who was willing to take the survey.

-What language was the survey given in? I am assuming Italian.

-The authors should consider providing the questions asked (or their translation to english) as supplemental materials that would allow the reader to assess potential bias.

-The authors should avoid the use of inflamatory terms such as "mutilation" that would bias the survey responder.

-The authors are making global conclusions based on a highly limited sampling of one locale within Italy. 

The revised manuscript is significantly improved, but still requires some relatively minor revisions, primarily of an editorial nature.  These are listed below.

General.  This reviewer continues to recommend that the authors include the questions asked as a supplemental file.  Similarly, since the population surveyed was a single town in a single region of Italy, the authors need to limit the interpretation of the results primarily to the locale and avoid extrapolating globally.

Line 26.  Words that appear in the title should not be repeated as keywords.

Line 39.  Delete “a guarantee of”.  Nothing is guaranteed.

Line 46.  Delete “After all, the”.

Line 55.  Delete “Furthermore,”.

Line 64.  What society are you referring to.  Local? Italy? Global?  Considering that the authors surveyed a single town in Sicily, they can only make statements about that society.

Lines 69-72.  Divide into two sentences.

Line 121-122.  Avoid one sentence paragraphs.

Lines 245-252.  Is this paragraph pertinent to the overall goal of the study to consumer preferences and attitudes?  Suggest that the paragraph be deleted since the authors did not ask about it in their survey.

Lines 259-263.  The only thing that the authors can state is that these are the consumer preferences in one town in Sicily.

References.  The authors need to review the journal’s style guidelines, and review their requirements for the capitalization of article titles.  The references lack consistency.

Author Response

As pointed out by the authors, there is a high likelihood of sampling bias based on who was willing to take the survey.

What language was the survey given in? I am assuming Italian.

Authors’ response: yes, it is italian.

The authors should consider providing the questions asked (or their translation to english) as supplemental materials that would allow the reader to assess potential bias.

Authors’ response: our questionnaire was attached as “supplementary matherial”.

The authors should avoid the use of inflamatory terms such as "mutilation" that would bias the survey responder.

Authors’ response: we are aware that this term might be conditioning the respondents' answers, but there is no synonym to the word "mutilation".

The authors are making global conclusions based on a highly limited sampling of one locale within Italy.

Authors’ response: We are aware that our results are limited to the area in which the survey was conducted. On the other hand, we specify that the reference population is the province of Messina, which has 218,187 subjects. For this reason, our sample can be considered a large sample by representation and is suitable to ensure the extensibility of the results.

Comments on the Quality of English Language

The revised manuscript is significantly improved, but still requires some relatively minor revisions, primarily of an editorial nature. These are listed below.

General. This reviewer continues to recommend that the authors include the questions asked as a supplemental file. Similarly, since the population surveyed was a single town in a single region of Italy, the authors need to limit the interpretation of the results primarily to the locale and avoid extrapolating globally.

 Authors’ response: We thank the referee for the valuable suggestion to improve the overall quality of our manuscript. In the revised version of our article, we specified in section 5 (Conclusion) that “The results of our survey can be traced to Southern Italy and, particularly, in the city of Messina (Sicily)”.

Line 26. Words that appear in the title should not be repeated as keywords.

Authors’ response: we have granted the referee's request and the previous keywords (consumer behavior; preferences; sustainable pork livestock; animal welfare) have been replaced with the following: purchasing habits, choices, sustainable livestock, animal welfare

Line 46. Delete “After all, the”.

Authors’ response Ok, we deleted the term.

Line 55. Delete “Furthermore,”.

Authors’ response Ok, we deleted the term.

Line 64. What society are you referring to. Local? Italy? Global? Considering that the authors surveyed a single town in Sicily, they can only make statements about that society.

Authors’ response : We added this sentence in the manuscript “in general worldwide and, especially in Italy”

Lines 69-72. Divide into two sentences.

Authors’ response: Ok, we divided into two sentences.

Line 121-122. Avoid one sentence paragraphs.

Authors’ response: Ok, thanks for your valuable advice

Lines 245-252. Is this paragraph pertinent to the overall goal of the study to consumer preferences and attitudes? Suggest that the paragraph be deleted since the authors did not ask about it in their survey.

Authors’ response: We are aware that COVID issues were not presented to respondents; however we cannot fail to highlight the role of Covid, which has certainly changed the consumer's eating habits.

Lines 259-263. The only thing that the authors can state is that these are the consumer preferences in one town in Sicily.

Authors’ response: We thank the referee for the valuable suggestion to improve the overall quality of our manuscript. In the revised version of our article, we specified in section 5 (Conclusion) that “The results of our survey can be traced to Southern Italy and, particularly, in the city of Messina (Sicily)”

References. The authors need to review the journal’s style guidelines, and review their requirements for the capitalization of article titles. The references lack consistency. 

Authors’ response: Thank you for your suggestion. We have made the requested changes

Round 2

Reviewer 2 Report

Dear Authors,

you have modified manuscript, not extensively but it is an obvious improvement. Some corrections are made, more on citing literature side, and less in the text. You have elaborated in the response to the reviewer, and that was done sufficiently. There are still some issues in the introduction, like stated below:

 Line 13, 21 and 22: term “sustainable” should be aligned with new term welfare-friendly; please revise abstract

Lines 31-42: first paragraph of the introduction was unchanged, only slight modifications in citing literature, but there are still unmatched literature conclusions and authors abstract. For instance, literature 5 (Welfare Health and Productivity in Commercial Pig Herds) does not mention any antibiotic use, but authors used it in their elaboration.

As stated before, preventive use of antibiotics is prohibited in EU, and this premises is still used in the introduction. Question is why authors persist on that formulation because presentation is more on laic level not on expert, especially in EU pig production. Please provide eligible evidence of misuse of antibiotics in EU, not speculations.

Furthermore, authors are discussing largely on antibiotics in first two paragraphs of the introduction but antibiotics were not investigated at all in their research. Authors are encouraged to modify these sections for better understanding of the topic.

Author Response

Line 13, 21 and 22: term “sustainable” should be aligned with new term welfare-friendly; please revise abstract.

Authors’ response: we have revised the abstract

Lines 31-42: first paragraph of the introduction was unchanged, only slight modifications in citing literature, but there are still unmatched literature conclusions and authors abstract. For instance, literature 5 (Welfare Health and Productivity in Commercial Pig Herds) does not mention any antibiotic use, but authors used it in their elaboration.

Authors’ response: we have changed the reference 5 and rephrased the sentence (lines 39-41).

As stated before, preventive use of antibiotics is prohibited in EU, and this premises is still used in the introduction. Question is why authors persist on that formulation because presentation is more on laic level not on expert, especially in EU pig production. Please provide eligible evidence of misuse of antibiotics in EU, not speculations.

Authors’ response: obviously, Regulation (EU) 2019/6 extendes the restrictions for the use of some antibiotics in animals to a complete ban on some antibiotics. It is true that "antimicrobial medicinal products shall not be applied routinely nor used to compensate for poor hygiene, inadequate animal husbandry or lack of care or to compensate for poor farm management" (art. 107, point 1), and "... shall not be used for prophylaxis" (where prophylaxis means the administration of a medicinal product to an animal or group of animals before clinical signs of a disease, in order to prevent the occurrence of disease or infection), but they can be used "... in exceptional cases, for the administration to an individual animal or a restricted number of animals when the risk of an infection or of an infectious disease is very high and the consequences are likely to be severe" (art. 107, point 3)

Furthermore, authors are discussing largely on antibiotics in first two paragraphs of the introduction but antibiotics were not investigated at all in their research. Authors are encouraged to modify these sections for better understanding of the topic.

Authors’ response: thank you for your suggestions. We modified the line 46-49. In these first two paragraphs we have wanted to create a link with the following matter in the manuscript  about One Health concept. In this framework we are discussing on antibiotic resistance that is one of the main challenges for the protection of humans, animals and environmental. Antimicrobial use in farm, in fact - such as we had to add in the text for better understanding of the topic- requires a global efforts by all involved actors in order to avoid global emergencies relating to their use'consquences.